

**Development of a three-dimensional variational assimilation**
**system for lidar profile data based on a size-resolved aerosol**
**model in WRF-Chem model v3.9.1 and its application in PM$_{2.5}$**
**forecasts across China**
Yanfei Liang[1,2], Zengliang Zang [1], Dong Liu[3], Peng Yan[4], Yiwen Hu[5], Yan Zhou[6], Wei You[1]
[1]Institute of Meteorology and Oceanography, National University of Defense Technology,
Nanjing, China.
[2]No.32145 Unit of PLA, Xinxiang, China
[3]Key Laboratory of Atmospheric Optics, Institute of Optics & Fine Mechanics, Chinese
Academy of Sciences, Hefei, China
[4]Meteorological Observation Center, Chinese Meteorological Administration, Beijing,
China
[5]Nanjing University of Information Science & Technology, Nanjing, China
[6]No.78127 Unit of PLA, Beijing, China

24 *Corresponding author: Wei You (ywlx_1987@163.com); Zengliang

(zzlqxxy@163.com)



## Abstract:

For the aerosol variables in the model for simulating aerosol interactions and chemistry (MOSAIC)-4bin chemical scheme in the Weather Research and Forecasting–Chemistry (WRF–Chem) model, this study presents an observation forward aerosol extinction coefficient (AEC) and aerosol mass concentration (AMC) operator and corresponding adjoint based on the interagency monitoring of protected visual environments (IMPROVE) equation, and then a three-dimensional variational (3-DVAR) data assimilation system (DA) is developed for lidar AECs and AMCs. DA experiments are conducted based on AEC profiles measured by five light detection and ranging (lidar) systems as well as mass concentration (MC) data measured at over 1,500 ground environmental monitoring stations across China for particulate matter 2.5 µm or less in diameter ($PM_{2.5}$) and PM between 2.5 and 10 µm in diameter ($PM_{10}$). An experiment comparing assimilated and without assimilated measurements finds the following. While only five lidars were available within the simulation region (approximately 2.33 million $km^2$ in size), assimilating lidar AEC data alone can effectively improve the accuracy of the initial field of the WRF–Chem as well as its forecast performance for $PM_{2.5}$MCs. Compared to the without assimilated experiment, DA reduces the root mean square error of surface $PM_{2.5}$MCs in the initial field of the model by 10.5 $µg/m^3$ (17.6%). Moreover, the positive effect resulting from the optimization of the initial field for AMCs can last for more than 24 h. By taking advantage of lidar aerosol vertical profile information and the near-surface PM MC observations, assimilating lidar AEC and surface $PM_{2.5}$ ($PM_{10}$) simultaneously can effectively integrate their observed information and generate a more accurate 3D aerosol analysis field.





## 1. Introduction

Aerosol data assimilation (DA) generates a three-dimensional (3D) gridded analysis field capable of describing the spatial distribution of aerosols by integrating numerical forecasts produced by an air quality model (AQM) and measured aerosol data. With integrated information from various sources, this analysis field can more accurately describe the 3D distribution pattern of aerosols (Carmichael et al., 2008; Benedetti et al., 2009; Sandu et al., 2011; Bannister, 2017). On the one hand, the analysis field generated by DA can be used to effectively study atmospheric aerosol transmission patterns by analyzing products of a certain time series and, on this basis, further examine the effects of aerosols on human health, the environment, the weather, and the climate (Baraskar et al., 2016). On the other hand, the analysis field can be used as the initial chemical conditions for an AQM. The forecast performance of the AQM for aerosols can then be enhanced by improving the accuracy of the initial chemical conditions (Wu et al., 2015).

Compared to meteorological and marine DA, aerosol DA techniques are still undeveloped. There is also a lack of variety when it comes to assimilable measured data, which mainly include conventional surface aerosol mass concentration (AMC) data and satellite-derived aerosol optical depth (AOD) data. Of these two types of data, surface AMC data directly provide mass concentration (MC) information for near-surface aerosols. The AOD is a measure of the total extinction effects of aerosols in the vertical atmospheric column. Thus, AOD data indirectly provide atmospheric-column concentration information of aerosols. Assimilating either of these two types of data can significantly improve the accuracy of the aerosol analysis field (Tombette et al., 2008; Niu et al., 2008; Schwartz et al., 2012; Jiang et al., 2013; Li et al., 2013; Saide et al., 2013; Yumimoto et al., 2015, 2016; Tang et al., 2017; Peng et al., 2017; Xia et al., 2019; Wang et al., 2020), but these two types of data are unable to provide vertical aerosol profiles. Consequently,





while these two types of data are abundant, have relatively high horizontal
resolutions, and cover a wide range of space, they play a very limited role in
optimizing the vertical structure of aerosols in the analysis field. To further
improve the accuracy of the vertical structure of aerosols, it is necessary to
assimilate measurements that contain vertical aerosol profile information.
Zang et al. (2016) assimilated aircraft-measured vertical concentration profiles
of aerosol components and found that while the profile data were limited in
quantity and covered a relatively small area, they could still significantly
improve the forecast accuracy of the AQM. Since direct observations of
concentration profiles require great amounts of labor and financial resources,
relatively few studies involving the acquisition and assimilation of this type of
data have been reported.

Light detection and ranging (lidar) can be used to capture

aerosol-backscattered laser signals at various heights. By inverting these
signals, the aerosol extinction coefficient (AEC) and aerosol backscattering
coefficient (ABC) can be determined, which indirectly provide vertical AMC
profile information (Fernald et al., 1984; Sugimoto et al., 2008, Raut et al.,
2009). Assimilating lidar aerosol data can help to improve the accuracy of the
vertical structure of aerosols in the analysis field (Sugimoto et al., 2009;
Tesche et al., 2007; Dilip et al., 2009; Young, S. A., and M. A. Vaughan, 2009;
Burton et al., 2010; Lolli et al., 2014; Chen et al., 2015). In addition, with the
increasing number of lidar stations and the development of lidar network
detection technology, there is great theoretical and application value to
studying lidar DA in order to generate more accurate 3D aerosol analysis
fields.

Compared to the assimilation of direct AMC measurements, the

assimilation of lidar AEC data faces myriad difficulties, of which establishing
an observation operator for the DA cost function is the most challenging. AEC
is the object of DA (i.e., observation variable), whereas the AMCs of various



types of aerosol variables in the AQM are to be optimized. To directly
determine optimal model aerosol variables by solving the DA cost function, it
is necessary to map the aerosol variables in the AQM to the observation space
by conducting a forward process on the observation operator (Kahnert et al.,
2008), corresponding to the calculation of the AMC from the AEC. In addition,
in 3D variational DA, it is also necessary to conduct the reverse process on the
observation operator when calculating the gradient of the cost function (Sandu
et al., 2011). The computational program for this adjoint process on the
observation operator relies on its forward process. The computational load and
the size of the program code increase nonlinearly with the complexity of the
forward process. Moreover, when it comes to aerosol variables, there are
many kinds of chemicals and particle-size bins. As a result, the chemical
model inherently involves a high computational load. Therefore, when using a
variational method to assimilate lidar data, it is necessary to take into
consideration both the accuracy and complexity of the observation operator.
Currently, there are three main methods that could be used to design
observation operators. (1) Directly using the Mie equation. Under the
assumption that aerosol particles are uniform and spherical, the Mie equation
can describe the scattering and extinction properties of aerosol particles of any
scale and with any chemical and physical parameters (Cheng et al., 2019).
However, since accurately solving the Mie equation involves a nonlinear
calculation process that contains iterations, it is extremely complicated to
implement, upgrade, and maintain the program for the reverse process on the
observation operator. In addition, due to the lack of reliable measurements of
essential aerosol parameters (e.g., complex refractive index, particle number
spectrum, and hygroscopicity), in practice it is necessary to introduce
assumptions about these parameters in DA schemes. This renders it difficult to
realize the high-accuracy advantage of DA schemes in practice. (2) Using the
community radiative transfer model (CRTM). This model is advantageous





because it gives the Jacobian term needed for the reverse process on the
observation operator when conducting its forward process, so that introducing
the CRTM to a DA scheme does not require separate numerical computational
programming for the reverse process on the observation operator (Liu and
Weng, 2006). DA schemes based on the CRTM have been applied in AOD
DA research and yielded excellent results (Liu et al., 2011). However, the
CRTM was developed for the Goddard Chemistry Aerosol Radiation and
Transport (GOCART) aerosol scheme in the Weather Research and
Forecasting–Chemistry (WRF–Chem) model. As a result, when applying the
CRTM to other AQMs and aerosol schemes, it is necessary to design
corresponding variable transformation interfaces (Cheng et al., 2019), which
will introduce additional errors. (3) Using the interagency monitoring of
protected visual environments (IMPROVE) equation. The IMPROVE
equation maps a relation between AMC and AEC (Lowenthal et al,. 2003;
Ryan et al,. 2005; Pitchford et al,. 2007; Gordon et al,. 2018). With relatively
high computational accuracy, this method has been used to evaluate model
performance and the extinction contributions of various aerosols (Kim et al.,
2006; Roy et al., 2007; Tao et al., 2009, 2012, 2014; Cao et al., 2012a, 2012b).
In addition, as its highest-order term is quadratic, the IMPROVE equation has
low nonlinearity. Therefore, using the IMPROVE equation to design an
observation operator can significantly reduce the complexity of the DA
program. To date, no operator design based on the IMPROVE equation and
subsequent variational lidar DA have been reported.
Some progress has been made in lidar DA. For example, Sekiyama et al.
(2010) used the Kalman filter DA method to assimilate ABC and AEC
profiles acquired by the Cloud-Aerosol Lidar and Infrared Pathfinder Satellite
Observations mission and applied the assimilated data to a global chemical
transport model. Wang et al. (2013, 2014a, and 2014b) studied the
assimilation of range-corrected lidar signals using the optimal interpolation



DA method and conducted an assimilation experiment based on data captured by 12 lidar positioned in the Mediterranean Basin and one lidar positioned on the French island of Corsica. They found that the improvement brought by DA to the forecast performance for PM$_{2.5}$ lasted for approximately 36 hours. However, in the above-mentioned studies, sequential DA methods were used, and there is no particular need to take into consideration the complexity of the observation operator. Cheng et al. (2019) assimilated lidar AEC profiles using a 3D variational DA method with an observation operator based on the CRTM, which was designed for the relatively simple GOCART dust aerosol scheme.

This study presents an observation operator and corresponding adjoint module developed for the AEC based on the IMPROVE equation. This observation operator module is introduced into the DA system developed by Li et al. (2013) and Zang et al. (2016) for the model for simulating aerosol interactions and chemistry (MOSAIC) aerosol scheme oriented to the WRF–Chem model. DA and forecast experiments are conducted based on data captured by five lidars (located in Beijing, Shijiazhuang, Taiyuan, Xuzhou, and Wuhu, respectively) as well as data collected at approximately 1,500 ground environmental monitoring stations for PM$_{2.5}$ and PM$_{10}$.

**2. Materials and Methods**

**2.1. AQM**

The WRF–Chem model version 3.9.1 was selected as the AQM. The model has 40 vertical layers between the surface and 50 hPa, with the resolution gradually decrease from the bottom up. The model domains are double-nested, and the second domain (D02) is centered at (114.57°E, 37.98°N) and has 175×166 grid points with a grid interval of 9 km. D02 covers the central and eastern regions of China (most of North China, northern Central China, northern East China, and eastern Northwest China) (Figure 1).





The MOSAIC_4bin aerosol scheme was adopted for simulations. This scheme
can describe eight aerosol types. For each aerosol type, there are four
particle-size bins (4bin). The following summarizes other physical and
chemical schemes used in this study: the carbon-bond mechanism version Z
(CBMZ) chemical reaction mechanism, the fast-J photolysis calculation
scheme, the rapid radiative transfer model for general circulation models
(RRTMG) shortwave radiation scheme, the RRTMG longwave radiation
scheme, the WRF single-moment 5-class microphysical scheme, the unified
Noah land-surface parameterization scheme, the Grell 3D ensemble cumulus
parameterization scheme, the Yonsei University planetary boundary layer
scheme, and the revised MM5 Monin–Obukhov near-surface layer scheme.

**2.2. Data**
AEC profiles used in this study were derived from data captured by five
lidars (positioned in Beijing, Shijiazhuang, Taiyuan, Xuzhou, and Wuhu,
respectively) between 0000 and 1200 Coordinated Universal Time (UTC) on
November 13, 2018 (Figure 1) at a wavelength of 532 nm. The temporal
resolution of the data measured by the lidars in Shijiazhuang, Taiyuan,
Xuzhou, and Wuhu is 1 min, i.e., data were captured and a vertical AEC
profile was derived every minute. The vertical resolution of these data is 7.5 m,
i.e., one AEC was determined in one profile 7.5m away from the next one.
The blind spot of these lidars is 100 m, i.e., these systems cannot effectively
capture AEC data between the height of 100 m and the surface. The temporal
and vertical resolution of the AEC profiles captured by the lidar in Beijing are
1 h and 15 m, respectively, and the blind spot of this lidar system is 210 m. To
improve the effects of DA, it is necessary to first perform quality control on
and preprocess the original AEC profiles. This will ensure that the data are of
relatively high temporal and spatial representativeness and that the Lidar data
match the numerical model in terms of temporal and spatial resolution.





Quality control mainly involves four steps. (1) Entire AEC profiles passing
through low clouds and AEC measurements in mid- and high-cloud regions
are eliminated. An AEC profile is deemed to pass through low clouds when
the AEC in the near-surface layer (below 150 m) is lower than $3,000 \times 10^{-6}$ $m^{-1}$
and there is an AEC higher than $5,000 \times 10^{-6}$ $m^{-1}$ below 800 m. AEC
measurements in mid- and high-cloud regions are determined as follows: If the
AEC in the near-surface layer (below 150 m) is lower than $3,000 \times 10^{-6}$ $m^{-1}$,
then measurements higher than $5,000 \times 10^{-6}$ $m^{-1}$ on the AEC profile are of
AECs in mid- and high-cloud regions. (2) AEC profile data are subjected to
maximum and minimum control. AEC measurements higher than $3,000 \times 10^{-6}$
$m^{-1}$ are each reassigned with a value of $3,000 \times 10^{-6}$ $m^{-1}$. AEC measurements
lower than $20 \times 10^{-6}$ $m^{-1}$ are eliminated. (3) Spatial continuity verification.
Valid data should be continuous within a certain continuous vertical space $L_{con}$
which is set to be 90 m in this study. Specifically, two metrics are used to
examine the spatial continuity of data. First, the profile with vertical resolution
$L_{res}$ is examined. After the first two steps of quality control, the remaining
number of data points ($N_{remain}$) within the $L_{con}$ should not be less than 1/3 the
total number of data points within the $L_{con}$ ($N_{total} = L_{con}/L_{res}$); otherwise, it is
considered that no valid data are available for the center of the $L_{con}$. Second,
the deviation of the valid data from the mean value of the data within the $L_{con}$
does not exceed 3times the standard deviation (SD). (4) Blind detection spot
verification. Data within the blind spot of a lidar are eliminated. In addition,
considering that lidar signals are relatively weak and AMCs are very low
above 5,000 m, data for the region above 5,000 m are also eliminated in this
study.

Preprocessing of quality control-treated AEC profiles involves two steps.

(1) Temporal and spatial smoothing. Profiles are subjected to moving
averaging over 30 m in the vertical direction. Temporally, AEC profiles are
also averaged over the past hour. (2) Data thinning. Only one valid data point





is selected for assimilation between two adjacent model layers in the vertical
direction. In this study, the nearest data below each model layer are selected
for assimilation. After processing, the number of assimilated AEC
measurements on each profile does not exceed 25.
PM$_{2.5}$ and PM$_{10}$ data (hereinafter referred to as PM data) used in this
study, including 1-h MC data collected at more than 1,500 ground
environmental monitoring stations, originated from the China National
Environmental Monitoring Center. Most of the monitoring stations are
distributed in cities in economically developed regions, including the Yangtze
River Delta, the Beijing–Tianjin–Hebei region, and the Pearl River Delta. Of
these monitoring stations, more than 790 are located within the D02 region
(Figure 1). DA experiments are performed in this study based on PM data
collected between 00:00 and 12:00 UTC on November 13, 2018. Subsequently,
forecasts for PM$_{2.5}$ from 12:00 UTC on November 13, 2018 to 12:00 UTC on
November 14, 2018 are produced. In addition, the effects of DA on the
forecast performance of the model are evaluated based on surface PM$_{2.5}$
measurements. To improve the effects of DA and the representativeness of the
evaluation metrics, the original PM data are subjected to quality-control and
preprocessing treatments. Quality control mainly involves two steps. (1)
Anomaly elimination. Measurements that remain unchanged over a continuous
period of 24 h are considered anomalous records and removed. (2) Maximum
and minimum control. PM$_{2.5}$MC measurements higher than 600 μg/m$^3$,
PM$_{10}$MC measurements higher than 1,200 μg/m$^3$, and PM MC measurements
less than 0 are considered anomalies and removed. During the DA and
verification processes, there may be multiple PM MC measurements for one
grid cell. To allow the measurements to represent the average PM MC within
a certain area, the PM data used for DA and verification are subjected to
grid-cell averaging. The PM data used for assimilation are averaged within
5×5 grid cells. Specifically, the PM data within the same 5×5 grid cell area are



first examined to determine their spatial consistency. Data greater than twice
the SD are removed. Then, the arithmetic mean of the data within the area is
calculated and assimilated. The $PM_{2.5}MC$ measurements used for verification
and model forecasts are averaged within 1×1 grid cells. Specifically, model
forecasts are first interpolated to the location of each ground environmental
monitoring station. Then, the arithmetic mean of the measured and forecasted
values within the same grid cell is calculated and used as a sample for
quantifying the evaluation metrics. The processed PM MC data for the D01
and D02 regions are all assimilated. Only the $PM_{2.5}MC$ data for the D02
region are used to evaluate the effects of DA. After the grid-cell averaging
treatment, approximately 190 data points in the D02 region are assimilated
each time.

**2.3. Basic theoretical DA model**


To mathematically achieve 3D variational DA, it is necessary to establish
an objective function to transform the DA problem to a problem of finding the
extreme value of the function. By calculating the extreme value of the function
using the variational method, an "optimal" analysis field will be obtained. The
following shows the mathematical form of such a function:

$$J(x) = \frac{1}{2}(x-x^b)^T B^{-1}(x-x^b) + \frac{1}{2}(Hx-y)^T R^{-1}(Hx-y) \qquad (1)$$

This function describes the sum of the distance between the analysis field
($x$) and the background field ($x^b$) and the distance between the analysis field ($x$)
and the observation field ($y$), with the background error covariance $B$ and the
observation error covariance $R$ as weights, respectively. In Equation (1), $x$ is
the control variable in the DA system, which is a one-dimensional (1D) vector
composed of aerosol variables at all the 3D grid cells in the DA analysis field;
$x^b$ is the background value (or best guess) of the control variable (as the
forecast level of AQM increases, model forecasts are generally used as





background fields); $B$ is the background error covariance; $y$ is the observation
variable, which is a 1D vector composed of all the measurements; $H$ is the
observation operator, which maps the control variable to the observation space
to ensure that the observation data can provide observation information for the
control variable even if they are not direct measurements of the control
variable; and $R$ is the observation error covariance. For simultaneous
assimilation of two or more types of observation data, the second term on the
right side of Equation (1) can be expanded to multiple terms, each of which
corresponds to one type of observation data. This will facilitate the
simultaneous assimilation of observation data from various sources.
**2.4. Control variables and B**
The MOSAIC_4bins aerosol scheme adopted in this study can describe
eight aerosol types, namely, black/elemental carbon (EC/BC), organic carbon
(OC), sulfates ($SO4^{2-}$), nitrates ($NO3^{-}$), ammonium salts ($NH4^{+}$), chlorides
($Cl^{-}$), sodium salts ($Na^{+}$), and other unclassified inorganic compounds (OIN).
There are four particle-size bins (4bin) for each aerosol type, namely,
0.039–0.1, 0.1–1.0, 1.0–2.5, and 2.5–10 µm. Thus, there are a total of 32
model variables that describe aerosols. Due to limitations of computer
memory and computational capacity, there cannot be an excessively large
number of control variables. In addition, fine ($PM_{2.5}$) and coarse ($PM_{2.5-10}$)
particles differ relatively significantly in AEC. Thus, two control variables,
namely, the sum of the first three particle-size bins (corresponding to fine
particles) and the fourth particle-size bin (corresponding to coarse particles),
are designed for each aerosol type, so that there are 16 control variables are
designed in this study for the DA scheme namely $EC_{2.5}$, $EC_{2.5-10}$, $OC_{2.5}$,
$OC_{2.5-10}$, $SO4_{2.5}$, $SO4_{2.5-10}$, $NO3_{2.5}$, $NO3_{2.5-10}$, $NH4_{2.5}$, $NH4_{2.5-10}$, $CL_{2.5}$, $CL_{2.5-10}$,
$NA_{2.5}$, $NA_{2.5-10}$, $OIN_{2.5}$, $OIN_{2.5-10}$.
Calculation associated with $B$ is burdened with two problems: (1) An





overly large scale of *B*. In this scheme, *B* contains $3.5 \times 10^{14}$ (= 16 (number of
control variables) × 175 × 166 × 40 (number of grid cells)) elements. Thus, it
is necessary to mathematically treat and approximately simplify *B* to facilitate
numerical calculations. Following the method used by Li et al. (2013) and
Zang et al., (2016), *B* is decomposed into a background-error SD matrix and a
background-error correlation coefficient matrix for calculations. (2) As the
true value of *B* is unknown, it is necessary to develop a reasonable statistical
method to estimate it. The National Meteorology Center (NMC) method
(Parrish and Derber, 1992) is employed in this study to statistically estimate *B*.
Specifically, the differences between 48h and 24h forecasts of control
variables are assumed to be a proxy of background error. Then, *B* is estimated
based on the covariance of the difference field, which is obtained by
producing continuous 24h and 48h forecasts for a month using the
WRF–Chem model.
**2.5. Observation forward operator and its ajoint**
The observation forward operator involves two steps of calculation. (1)
The control variables at each grid cell are mapped to the observation space,
i.e., the control variables are mapped to AEC values (or $PM_{2.5}$ and $PM_{10}$MCs).
(2) The mapped values at the eight vertices of the model grid cell associated
with the observation data are interpolated using the inverse distance-weighted
method to the observation location. Here we only describe the first step of the
observation operators which is different for different observation data.
The AEC observation operator is based on the IMPROVE equation. The
following shows the specific form of the IMPROVE equation:

$$
\begin{aligned}
Ext=\ &3.025 \times fs(RH) \times [Small\ Sulfate]+ \\
&6.6 \times fl(RH) \times [Large\ Sulfate]+ \\
&3.096 \times fs(RH) \times [Small\ Nitrate]+ \\
&6.579 \times fl(RH) \times [Large\ Nitrate]+ \\
&5.04 \times [Small\ Organic\ Mass]+ \\
&10.98 \times [Large\ Organic\ Mass]+ \\
&10.0 \times [Elemental\ Carbon]+
\end{aligned} \tag{2}
$$





$$1.0 \times [Fine\ Soil] +$$
$$1.7 \times fss(RH) \times [Sea\ Salt] +$$
$$1.0 \times [Coarse\ Mass]$$

The left side of Equation (2) is the AEC value *Ext* (unit: $10^{-6}$ m$^{-1}$). The
variables in the brackets on the right side of Equation (2) are combinations of
the 16 control variables (unit: μg/m$^3$). The coefficient variables $fs(RH)$, $fl(RH)$,
and $fss(RH)$ reflect the effects of hygroscopicity of fine, coarse, and sea-salt
aerosols in various relative humidity (HR) conditions on extinction efficiency,
respectively. The values of parameters given by Gordon et al. (2018) are used
in this study. The variables (in the square brackets) at each grid cell are
obtained by combining the 16 control variables using the following method:
$$Sulfate = SO4_{2.5} + \alpha \times NH4_{2.5}.$$
The principle for determining α is: $NH4_{2.5}$ is preferentially allocated to
$SO4_{2.5}$, and the remaining $NH4_{2.5}$ is allocated to $NO3_{2.5}$.
$$[Small\ \ Sulfate] = \begin{cases} 0 & , Sulfate\ >= 20 \\ (1 - \dfrac{Sulfate}{20}) \times Sulfate & , Sulfate\ < 20 \end{cases}$$
$[Large\ Sulfate] = Sulfate - [Small\ Sulfate]$
$Nitrate = NO3_{2.5} + (1 - \alpha)(NH4_{2.5})$
$$[Small\ \ Nitrate] = \begin{cases} 0 & , Nitrate >= 20 \\ (1 - \dfrac{Nitrate}{20}) \times Nitrate & , Nitrate < 20 \end{cases}$$   (3)
$[Large\ Nitrate] = Nitrate - [Small\ Nitrate]$
$[Organic\ Mass] = OC_{2.5}$
$$[Small\ \ Organic\ \ Mass] = \begin{cases} 0 & ,[Organic\ \ Mass] >= 20 \\ (1 - \dfrac{[Organic\ \ Mass]}{20}) \times [Organic\ \ Mass] & ,[Organic\ \ Mass] < 20 \end{cases}$$
$[Large\ Organic\ Mass] = [Organic\ Mass] - [Small\ Organic\ Mass]$
$[Elemental\ Carbon] = EC_{2.5}$
$[Fine\ Soil] = OIN_{2.5}$
$[Sea\ Salt] = CL_{2.5} + NA_{2.5}$
$[Coarse\ Mass] = SO4_{2.5\text{-}10} + NO3_{2.5\text{-}10} + NH4_{2.5\text{-}10} + OC_{2.5\text{-}10} +$
$\qquad\qquad EC_{2.5\text{-}10} + CL_{2.5\text{-}10} + NA_{2.5\text{-}10} + OIN_{2.5\text{-}10}$
The observation operator for each of PM$_{2.5}$ and PM$_{10}$ is the sum of
control variables in the corresponding particle-size bin, i.e.,
$PM_{2.5} = SO4_{2.5} + NO3_{2.5} + NH4_{2.5} + OC_{2.5} + EC_{2.5} + CL_{2.5} + NA_{2.5} + OIN_{2.5}$   (4)
$PM_{10} = SO4_{2.5} + NO3_{2.5} + NH4_{2.5} + OC_{2.5} + EC_{2.5} + CL_{2.5} + NA_{2.5} + OIN_{2.5} +$





$SO4_{2.5-10}+NO3_{2.5-10}+NH4_{2.5-10}+OC_{2.5-10}+EC_{2.5-10}+CL_{2.5-10}+NA_{2.5-10}+OIN_{2.5-10}$ (5)

The corresponding adjoint operators for PM and AEC are developed and
passed the adjoint sensitivity test. The adjoint test method please refer Zou et
al.(1997).
**2.6. DA and forecast experimental design and verification analysis method**
To analyze the effects of DA on aerosol analysis and forecasts, one
control experiment with unassimilated data and three DA experiments are
designed for a pollution event that occurred from November 13 to 14, 2018
(Table 1). In the control experiment, no chemical observation data are
assimilated. Forecasts are produced for a 36-h period, starting at 0000 UTC on
November 13, 2018. In the DA experiments, hourly aerosol data for the period
0000–1200 UTC on November 13, 2018 are assimilated. Then with the
analysis field obtained from DA as the initial chemical field, forecasts are
performed for a 24-h period starting at 1200 UTC on November 13, 2018. The
period 0000–1200 UTC on November 13, 2018 is set as DA period. For the
DA period, the first DA (0000 UTC on November 13, 2018) is performed with
the initial field of the control experiment as the background field. By
assimilating the observation data for 0000 UTC on November 13, 2018, a DA
analysis field is generated for this time point. With this DA analysis field as
the initial field at 0000 UTC, November 13, 2018 in the DA experiment, 1-h
forecasts are produced. And the forecasts produced for 0100 UTC, November
13, 2018 are used as the background field for the second DA. The process is
repeated until 13 assimilation cycles are completed. Thus, a DA analysis field
for 1200 UTC, November 13, 2018 is generated. The period from 1200 UTC,
November 13, 2018 to 1200 UTC, November 14, 2018 is selected to compare
model forecasts. The effects of DA on forecast performance can be analyzed
by comparing the DA and control experiments in terms of forecast
performance. The three DA experiments differ in assimilated data. In the first
DA experiment (DA_PM), PM data alone are assimilated. In the second DA





experiment (DA_Ext), lidar data alone are assimilated. In the third DA
experiment (DA_PM_Ext), PM and lidar data are assimilated simultaneously.
Furthermore, 0.25°×0.25° 6-h reanalysis data provided by the U.S. National
Centers for Environmental Prediction (NCEP) are used as the meteorological
field of the model.

Two metrics, namely, regional mean and root-mean-square error (RMSE),

are used to evaluate simulation and forecast accuracy for $PM_{2.5}$MCs in the
experiments. The closer the mean of simulated values is to the mean of
measurements and the smaller the RMSE is, the higher the performance is. Let
$M_i$, $O_i$, $N$, $\overline{M}$, and $\overline{O}$  be the simulated value sample, the measured value
sample, the number of samples, the mean of simulated values, and the mean of
measurements, respectively. The following summarizes the equation for
calculating each metric:
$$\overline{M} = \frac{1}{N}\sum_{i=1}^{N} M_i \qquad (6)$$

$$\overline{O} = \frac{1}{N}\sum_{i=1}^{N} O_i \qquad (7)$$

$$\mathrm{RMSE} = \sqrt{\frac{1}{N}\sum_{i=1}^{N}(M_i - O_i)^2} \qquad (8)$$

**3. Results**
**3.1. SD and vertical correlation coefficient of the background error**
**(BESD and BEVCC)**

Under the same conditions, the larger the BESD is, the greater the

increment caused by DA is. Therefore, the structural pattern of the BESD will
significantly affect the distribution pattern of the DA increment field. Figure 2
shows the vertical BESD profiles of the 16 control variables. As demonstrated
in Figure 2, the BESD differs relatively significantly between control variables.
The seven control variables with the largest BESDs below the height of 1,000
m (corresponding to the 22nd layer of the model) in descending order of BESD
are: $OIN_{2.5-10}$, $NO3_{2.5}$, $OIN_{2.5}$, $NH4_{2.5}$, $SO4_{2.5}$, $OC_{2.5}$, and $EC_{2.5}$. As height





increases, the BESD of each control variable decreases. The rates of increase
are the highest above the boundary layers at heights of 1,000–2,000 m
(corresponding to the $20^{th}$–$25^{th}$ layers of the model).
The BEVCC matrix can affect the vertical transference range of
observation information. Even the PM data are only available at surface, there
will still be DA increments of PM in near-surface in-air after PM DA.
Similarly, even no near-surface lidar data are available, assimilating lidar data
can still correct the surface $PM_{2.5}MC$ distribution. Figure 3 shows the BEVCC
matrices of six control variables with relatively large BESDs ($OIN_{2.5-10}$,
$NO3_{2.5}$, $OIN_{2.5}$, $NH4_{2.5}$, $SO4_{2.5}$, and $OC_{2.5}$). As demonstrated in Figure 3, the
BEVCCs of the control variables share certain common characteristics. The
correlation decreases as the interlayer spacing of the model increases. Each
in-air layer is positively correlated with the surface layer, though the
correlation decreases as height increases. $OIN_{2.5-10}$ has a significantly weaker
vertical correlation than the other variables. For $OIN_{2.5-10}$, the correlation
coefficient between the surface and $10^{th}$ layers is 0.34, compared with
0.49–0.51 for other variables. This is mainly because coarse particles settle
faster than fine particles and are concentrated near the surface in larger
quantities.
**3.2. Analysis of the pollution process**
Figure 4 shows the evolutionary process of surface $PM_{2.5}MC$
measurements and the NCEP reanalysis surface wind field in the D02 region
for the period from 0000 UTC, November 13, 2018 to 1200 UTC, November
14, 2018 (the time interval between Figure 4a, b, c, and d is 12 h). As
demonstrated in Figure 4a, at 0000 UTC on November 13, 2018, the D02
region was predominantly controlled by a high-pressure circulation centered
over Zibo in central Shandong province. There was a clockwise wind field
around the high-pressure center. There were northerlies (easterlies) east (south)





of the high-pressure center, bringing clean air over the sea landward. As a
result, $PM_{2.5}$MCs in East China were relatively low. For example, the mean
$PM_{2.5}$MC measured at the ground environmental monitoring stations in
Nanjing, Jiangsu province, was 41.8 μg/m³. There were relatively slow
southerlies west and northwest of the high-pressure center. This led to
favorable conditions for pollutant accumulation east of the Taihang Mountains
and south of the Yan Mountains. As a result, North China was relatively
heavily polluted by $PM_{2.5}$. For example, the mean $PM_{2.5}$MCs in Beijing and
Shijiazhuang, Hebei Province, were 122.7 and 149.3 μg/m³, respectively. In
addition, within the region, there was also a northeast–southwest-trending cold
front near Buyant-Ovoo–Bayan-Ovoo in Mongolia. As time passed (Figure 4b,
c, and d), the high-pressure center gradually moved northeastward and had
reached near the eastern boundary of the region by 1200 UTC, November 14,
2018 (Figure 4d). The cold front gradually moved southeastward and had
reached the Chaoyang–Beijing–Taiyuan–Xi'an line by 1200 UTC, November
14, 2018 (Figure 4d). As the high-pressure center and the cold front moved,
the level of pollution in North China continued to rise, and pollution gradually
expanded northeastward (Chaoyang, Liaoning Province), southward
(Zhengzhou, Henan Province), and westward (Taiyuan, Shanxi Province). Due
to the dual action of the advective transport by easterlies and the narrow
terrain, the level of pollution gradually increased in the Wei and Yellow River
Valleys east of Xi'an, Shaanxi Province. Thanks to good dispersion conditions,
$PM_{2.5}$MCs decreased considerably upon the passing of the cold front. There
were no significant changes in $PM_{2.5}$MCs in East China, owing to the
continuous impact of sea winds.
**3.3. Analysis of the direct affects of DA**

Figure 5 shows the AEC profiles captured at four lidar stations at 0000

UTC, November 13, 2018 as well as the corresponding AEC profiles in the





analysis fields of the control and DA experiments and the simulated RH
profiles. The first DA is performed for 0000 UTC, November 13, 2018. The
results of the control experiment are used as the background field in the three
DA experiments. Figure 5a, b, c, and d show the results for Beijing,
Shijiazhuang, Taiyuan, and Wuhu, respectively. As demonstrated by the RH
profiles (brown lines) in Figure 5, the RH in air below 1 km is basically
consistent with the surface RH. Thus, vertical changes in AEC values below 1
km are relatively insignificantly affected by RH, and the AEC profiles can
describe vertical changes in $PM_{2.5}MC$ profiles. A comparison of lidar AEC
profiles (black lines) and those obtained from the control experiment (blue
lines) finds that AEC values obtained from the control experiment are
relatively underestimated for Shijiazhuang (Figure 5b) and Taiyuan (Figure
5c), particularly near the height of 100 m (starting height for lidar data). In
comparison, AEC values for Wuhu (Figure 5d) obtained from the control
experiment are higher than the lidar measurements, while the AEC profile for
Beijing (Figure 5a) obtained from the control experiment is in relatively good
agreement with the lidar AEC profile. The AEC values for the Beijing (Figure
5a), Taiyuan (Figure 5c), and Wuhu (Figure 5d) stations obtained from the
DA_PM experiment (green lines) based on assimilated surface PM MC
measurements are lower than those obtained from the control experiment. This
is because the surface PM MC measurements used in the control experiment
for these three stations are relatively high. As a result of the BEVCC (Figure
3), PM DA will reduce the AEC values in lower in-air layers while reducing
surface PM MCs. In the DA_PM experiment, the adjustment made to the AEC
profiles for Beijing and Wuhu is, overall, positive, but the adjustment made to
the AEC profile for Taiyuan increases the underestimation of in-air AEC
values.
Compared to the DA_PM experiment, vertical aerosol distribution
patterns obtained from the DA_Ext experiment (purple lines) are more finely





adjusted. For example, the AEC values obtained from the DA_Ext experiment
for the Taiyuan station (Figure 5c) for the heights of approximately 100 and
700 m are significantly higher than those obtained from the DA_PM
experiment and are consistent with those on the lidar AEC profile (black line).
The AEC profile obtained from the DA_Ext experiment for the Wuhu station
(Figure 5d) is very close to the lidar AEC profile. This suggests that the AEC
observation operator designed based on the IMPROVE equation effectively
facilitates 3D variational assimilation of lidar AEC data. In addition, due to
the BEVCC (Figure 3), lidar DA will increase (decrease) surface PM MCs
while increasing (decreasing) in-air PM MCs. This surface PM MC
adjustment effectively corrects the overestimation of surface $PM_{2.5}$MCs in
Beijing and Wuhu in the control experiment but increases the overestimation
of surface $PM_{2.5}$MCs in Taiyuan.

The in-air AEC profiles obtained from the DA_PM_Ext experiment (red

lines) for the four cities almost coincide with those obtained from the DA_Ext
experiments above 400 m. The near-surface AEC values obtained from
DA_PM_Ext experiment for Beijing (Figure 5a) almost coincide with those
obtained from the DA_PM experiment. The near-surface AEC values obtained
from DA_PM_Ext experiment for Taiyuan (Figure 5c) are between those
obtained from the DA_PM and DA_Ext experiments. The near-surface AEC
values obtained from DA_PM_Ext experiment for Wuhu (Figure 5d) are
lower than those from the DA_PM and DA_Ext experiments. This suggests
that simultaneously assimilating two types of data can fully integrate their
observation information and reflect their respective advantages and, on this
basis, generate the most accurate analysis field.

Figure 6 shows the AEC profiles at 1200 UTC, November 13, 2018

measured at four lidar stations as well as the corresponding AEC profiles
obtained from the control experiment and the background and analysis fields
of the DA experiments. A total of 13 DA cycles are performed for the period



00:00–12:00 UTC, November 13, 2018. The time of 1200 UTC, November 13,
2018 is the last time point of the DA period and the starting time point of the
forecast period. The background field for each of the three DA experiments is
generated during the continuous DA period, whereas the results of the control
experiment are obtained by forecasting starting at 0000 UTC, November 13,
2018. As a result, there is a relatively significant difference between the
background fields of the three DA experiments and the results of the control
experiment.
As demonstrated in Figure 6, the results of the DA_PM experiment
(green lines) show significant $PM_{2.5}$MC increments below 1 km. The problem
of near-surface overestimation for the four cities in the control experiment is
corrected in the DA_PM experiment. This suggests that the DA yields a
positive effect. However, compared to those from the control experiment, the
AEC values obtained from the DA_PM experiment for Taiyuan at heights of
120–400 m (Figure 6c) and Wuhu above 400 m (Figure 6d) are even more
underestimated, suggests that the DA yields a negative effect here. It is worth
noting that there are very small direct DA increments (i.e., the differences
between the solid and dotted green lines) generated in the DA_PM experiment
at this time point. This means that for surface PM DA, a DA period of 11 h or
less is sufficient to effectively adjust aerosol distribution. This is because
aerosols are primarily concentrated near the surface and surface PM data cover
a wide area and have a high spatial resolution, so surface PM data measured at
a few time points contain the main aerosol distribution information for the
whole region.
Compared to the DA_PM experiment, the DA_Ext experiment (purple
lines) reflects the advantages of AEC DA in adjusting vertical aerosol
distribution. The problem of overestimation for Beijing above 300 m (Figure
6a), Taiyuan above 600 m (Figure 6c), and Wuhu below 400 m (Figure 6d) in
the control experiment is effectively corrected in the DA_PM experiment. In





addition, the results of the DA_Ext experiment reflect the rapid decrease in
PM$_{2.5}$MC with a height below 1 km in Beijing (Figure 6a) and the presence of
a maximum-PM$_{2.5}$MC layer at the height of 1.3 km in Wuhu (Figure 6d).
However, the near-surface overestimation for Taiyuan (Figure 6c) is increased
in the DA_Ext experiment. Moreover, the direct DA increments (i.e., the
differences between the solid and dotted purple lines) generated in the
DA_Ext experiment at this time point remain notable. This suggests that the
error of the background field at each lidar station at 1200 UTC on November
13, 2018 remains relatively large, even after a continuous DA period of 12 h.
To improve the effects of DA, it is necessary to increase the length of the
continuous DA period. This may be because there are few lidars and the lidars
are relatively far apart from one another. As a result, the simulation error for
the region upstream of the lidar is difficult to be corrected through DA and
will affect the lidar locate under advection at the next time point.

606  The AEC profiles obtained from the DA_PM_Ext experiment (solid red

lines) compared to the other two DA experiments show that the problem of
overestimation for Beijing above 400 m (Figure 6a), Shijiazhuang above 300
m (Figure 6b), and Wuhu in the near-surface layer(Figure 6d) in the control
experiment is considerably corrected in the DA_PM_Ext experiment. The
results of the DA_PM_Ext experiment reflect the advantage of simultaneously
assimilating two types of data in integrating their observation information.
This finding is consistent with Figure 5.

614  Figure 7 shows the surface PM$_{2.5}$MCs measured at 1200, November 13,

2018 as well as the corresponding initial field of the control experiment and its
error and the distribution of differences between the initial fields of the control
and DA experiments. As demonstrated in Figure 7a and 7b, the simulation
results obtained from the control experiment show that PM$_{2.5}$MCs are
relatively high in North China. In particular, there is a heavily polluted zone in
the Beijing–Shijiazhuang–Zhengzhou region, and PM$_{2.5}$MCs are relatively low





in the region surrounding North China. However, as demonstrated in Figure 7c,
$PM_{2.5}$MCs obtained from the control experiment are overestimated for most
regions. In particular, $PM_{2.5}$MCsobtained from the control experiment are
relatively highly overestimated for the Beijing–Shijiazhuang–Zhengzhou
region. In comparison, $PM_{2.5}$MCs obtained from the control experiment are
underestimated for the region near Chaoyang, Liaoning Province. The
distribution trends in Figure 7c and d are relatively consistent. This indicates
that the overestimation for most regions and the underestimation for some
regions in the initial field of the control experiment are corrected by PM DA.
As a result, the analysis field of the DA_PM experiment is closer to the
measurements.
A comparison of Figure 7c and 7e finds that significant DA increments
are generated in the DA_Ext experiment in the regions surrounding the five
lidar stations and the regions downstream of the wind field (Figure 4). Certain
DA increments are also present in regions far away from the lidar stations.
This indicates that long-term continuous assimilation of lidar measurements
can affect a relatively large area. Overall, AEC DA (from the DA_Ext
experiment) corrects the overestimation for most regions and the
underestimation for some regions in the initial field of the control experiment.
However, DA increments on the surface generated in the DA_Ext experiment
are smaller than those generated in the DA_PM experiment in terms of
horizontal spatial range and magnitude. This is mainly because there are
relatively few lidars and these lidars cover a limited spatial area. It is worth
noting that AEC DA yields a negative effect for northern Beijing and the
region around Taiyuan. For northern Beijing, the negative effect results
primarily from the notable overestimation for the location of the Beijing lidar
station, whereas the overestimation for northern Beijing is relatively low, and
there is even an underestimation for the locations of some individual stations
north of Beijing (Figure 7c). As a result, a negative effect is generated after





the DA increment at the location of the lidar station is transferred to northern
Beijing. For Taiyuan, the cause of the negative effect is similar to that seen in
Figure 5. According to the lidar measurements, the observation background
error at the height of approximately 100 m in the control experiment is
positive, i.e., there is an underestimation. However, according to the surface
PM MC measurements, the surface observation background error in the
control experiment is negative, i.e., there is an overestimation. Due to the
impact of the BEVCC, lidar DA will increase surface PM MCs while
increasing in-air PM MCs. This will lead to an increased overestimation of
surface PM MCs. There are two reasons for the presence of opposite in-air and
surface observation background errors. On the one hand, the simulation error
of the model is nonuniform in the vertical direction. On the other hand, the
opposite errors may be because the observation information in the lidar and
PM data was obtained from air parcels differing relatively significantly in
$PM_{2.5}$MCs in the horizontal direction. $PM_{2.5}$MCs measured at 1200 UTC,
November 13, 2018 at three ground environmental monitoring stations within
6 km of the Taiyuan lidar station were 80.0, 137.0, and 146.0μg/m3,
respectively. As also demonstrated in Figure 7a, $PM_{2.5}$MC measurements were
relatively low at most stations near Taiyuan (blue) but very high at two
stations (red). A similar phenomenon can be observed for the measurements
taken between 0000 and 1100 UTC. This suggests a relatively large horizontal
$PM_{2.5}$MC gradient near Taiyuan. The lidar and ground environmental
monitoring stations were situated at different locations. This led to a relatively
significant difference in $PM_{2.5}$MC data acquired in air parcels at the lidar and
ground environmental monitoring stations. All of this suggests that particular
attention should be paid to the horizontal spatial representativeness of lidar
data during the DA process.

A comparison of the results of the three DA experiments finds that the

results of the DA_PM_Ext experiment are in relatively good agreement with





those of the DA_PM experiment. This is mainly because the PM data are far
greater than the lidar data in terms of quantity and spatial coverage. As a result,
the DA increments in surface $PM_{2.5}$ concentrations originate primarily from
the observation information in the PM data. However, by analyzing Figures 5
and 6, it can be reasonably inferred that as height increases, the analysis field
of the DA_PM_Ext experiment will include more observation information in
the lidar data and thereby more accurately reflect the 3D spatial distribution
pattern of aerosols.
**3.4. Effects of DA on the forecast performance for surface $PM_{2.5}$MCs**

In this section, the effects of DA on forecast performance for aerosols are

evaluated based on approximately 430 surface $PM_{2.5}$MC measurements that
cover most of the D02 region.

Figure 8 shows the trend of the variation in the regional mean $PM_{2.5}$MC

with time in each of the four experiments. As demonstrated in Figure 8, the
variation in $PM_{2.5}$MC measurements (black line) exhibits a notable diurnal
pattern. Two notable minimum $PM_{2.5}$MCs (69.1 and 77.9 μg/m³) appeared at
0800 UTC (1600 local time), November 13, 2018 and 0800 UTC (1600 local
time), November 14, 2018, respectively. High $PM_{2.5}$MCs appeared between
1300 UTC, November 13, 2018 and 0200 UTC, November 14, 2018 (from
night to morning). The maximum $PM_{2.5}$MC was 96.0 μg/m³. However, there
was a relative minimum $PM_{2.5}$MC (87.0 μg/m³) appearing at 2200 UTC,
November 13, 2018 (around dawn local time) during the high-$PM_{2.5}$-MC
period. Comparing the control experiment with the measurements finds that
the experiment simulates the periodic variation pattern of the mean $PM_{2.5}$MC
(solid blue line). However, $PM_{2.5}$MCs obtained from the control experiment
are significantly overestimated for the whole forecast period. The $PM_{2.5}$MC
obtained from the control experiment for the initial time point (1200 UTC,
November 13, 2018) is overestimated by 36.3 μg/m³ (39.3%).



The overestimation in the control experiment is significantly reduced in
the DA_PM experiment (green line, which, partially, almost coincides with
the red line). The mean $PM_{2.5}$MC obtained from the DA_PM experiment for
1200 UTC, November 13, 2018 (91.4 μg/m³) is lower than that obtained from
the control experiment (128.6 μg/m³) by 37.2 μg/m³ (28.9%)and is closer to
the measurement (92.3 μg/m³). As a result of the decrease in the MC level in
the initial field, the $PM_{2.5}$MC forecasts obtained from the DA_PM experiment
are significantly lower than those obtained from the control experiment for the
whole forecast period. This suggests that the overestimation of the initial field
is the primary cause of the overestimated forecasts obtained from the control
experiment. In addition, DA can improve forecast results over a long time by
optimizing the initial field. In the DA_PM experiment, the effects of PM DA
last for more than 24 h. As demonstrated by the results of the DA_Ext
experiment (purple line), while there are only five lidars within the region,
AEC DA can still significantly correct the overestimation error of the initial
field and improve forecast performance. Compared to those in the DA_PM
experiment, the DA increments generated in the DA_Ext experiment are
relatively small and affect forecast results for a relatively short time
(approximately 21 h). This is mainly a result of the relatively small number of
lidars. There was no significant difference between the results of the
DA_PM_Ext (red line) and DA_PM (green line) experiments at surface. This
suggests that after surface PM DA, lidar DA relatively insignificantly affects
surface $PM_{2.5}$MCs. This happens for two reasons. On the one hand, similar to
the analysis of Figure 7f, of the two types of assimilated data, the proportion
of PM data is far greater than that of lidar data. On the other hand, after
surface PM DA, lidar DA affects surface aerosol forecasts mainly by adjusting
in-air AMCs and, on this basis, indirectly affects surface AMC forecasts by
processes such as settling. However, in this simulation process, surface AMC
remains at relatively high levels. Moreover, due to the relatively stable





meteorological conditions and weak vertical air movement in the simulation
region, particularly the heavily polluted zone, the indirect effects of lidar DA
are far smaller than the direct effects of PM DA on surface AMCs.
Figure 9 shows the variation in the RMSE of surface $PM_{2.5}MC$ forecasts
with time. A comparison of the RMSEs from the control experiment (blue line)
in Figure 9 and the mean $PM_{2.5}MCs$ obtained from the control experiment in
Figure 8 finds that the RMSEs for simulations and forecasts are relatively
large (small) at a relatively high (low) aerosol pollution level. As
demonstrated in Figure 9, the RMSE in the control experiment for the initial
time point (1200 UTC, November 13, 2018) of the forecast period is 59.6
$\mu g/m^3$. Throughout the forecast period, the RMSE fluctuates between 44.5 and
67.1 $\mu g/m^3$, instead of linearly increasing or decreasing. The RMSEs from the
DA_PM (green line), DA_Ext (purple line), and DA_PM_Ext (red line)
experiments for the initial time point are 21.0, 49.1, and 21.2 $\mu g/m^3$,
respectively, which are 38.6 (64.8%), 10.5 (17.6%), and 38.4 (64.4%) $\mu g/m^3$
lower than that for the control experiment. This suggests that the error of the
initial field is reduced in each of the three DA experiments. Thanks to an
optimized initial field, the RMSE of the forecasts produced in each of the DA
experiments is lower than that for the forecasts produced in the control
experiment. The RMSEs of the forecasts produced in the Da_PM, Da_Ext,
and DA_PM_Ext experiments for the 24th forecast hour are 6.1 (11.8%), 1.5
(2.9%), and 6.5 (12.6%) $\mu g/m^3$ smaller than that of the forecast produced in
the control experiment, respectively. This suggests that the optimization of the
initial field has a lasting (more than 24 h in all cases) positive effect on model
forecasts. It is worth noting that while there are very few lidar stations, the
results of the DA_Ext experiment are still better than those of the control
experiment, and the results of the DA_PM_Ext experiment are also slightly
better than those of the DA_PM experiment. This indicates that even in
relatively low quantities, lidar data still improve the forecast performance of



the model. As lidar data become increasingly rich and provide more vertical and horizontal aerosol distribution information in future, lidar DA will further improve $PM_{2.5}$MC forecasts.

## 4. Discussion

In Figure 7e, the relatively large AMC gradient in Taiyuan leads to opposite in-air and surface observation background errors , which lead to a negative effect of lidar DA for the surface. This suggests that the spatial representativeness of lidar data relatively significantly affects the impact of lidar AEC DA. In addition, the vertical resolution of lidar data (smaller than 15 m) is far smaller than the spacing between adjacent height layers of the model. As a result, the representative spatial scale of original lidar data does not match the resolution of the model. To improve the horizontal spatial representativeness of the lidar data, the lidar data of the past hour are averaged in this experiment as the lidar AEC profile for the time point. The vertical spatial representativeness of the data is improved by smoothing over 30 m in the vertical direction. However, the time-averaged lidar data represent observation information for a certain area downstream of the wind field. This representativeness error needs addressing in subsequent studies. Moreover, selection of a time averaging window length and a vertical smoothing length also requires further investigation.

For the Beijing region in Figure 7e, as a result of the relatively significant difference between the simulation error for the region downstream of the wind field and that for the location of the lidar station, the downstream transference of lidar DA increments will cause a negative effect in the continuous DA process. The most direct and effective measure for addressing this problem is to increase the number of lidars and the coverage of lidar network. This measure will ensure that the simulation error for the simulation region will be more comprehensively captured. However, lidar detection requires great





amounts of labor and financial resources. Therefore, it is difficult to arrange lidar stations as densely as ground environmental monitoring stations. A relatively feasible method is to set a relatively small number of lidars in regions with relatively small spatial changes in the simulation error and set dense lidars in regions with significant spatial changes in the simulation error. This will make it possible to use the limited number of lidars to capture more useful information. Thus, studying the temporal and spatial distribution pattern of model simulation errors can provide a useful reference for future arrangement and planning of lidar stations. This merits further investigation.

The AEC observation operator used in this study is designed based on the IMPROVE equation. The parameters of the IMPROVE equation, such as hygroscopicity coefficient, are directly set to values reported in previous studies. On the one hand, datasets used in previous studies were measured in specific regions. To date, no quantitative comparative analysis has yet to be performed to determine whether the extinction properties of aerosols differ between regions. Therefore, there is some uncertainty in the applicability of the IMPROVE equation. On the other hand, the values of the coefficients in the IMPROVE equation are determined by extensive statistical analysis of data. This dictates that these coefficients represent average levels under certain pollution and humidity conditions. There may be a certain error in these coefficients when applied to a specific observation event. This error will accumulate and amplify during the calculation of the forward and reverse processes of the observation operator, resulting in a negative effect of DA. Hence, how to effectively evaluate the applicability of the IMPROVE equation and more accurately adjust its coefficients is another issue that needs addressing.

## 5. Conclusions





In this study, an observation operator and its adjoint is designed based on
the IMPROVE equation to facilitate 3-DVAR assimilation of AEC data and a
3-DVAR DA system is developed for lidar AEC data and surface AMC data
for the MOSAIC-4bin chemical scheme in the WRF–Chem model. Three DA
experiments (i.e., a $PM_{2.5}(PM_{10})$ DA experiment, a lidar AEC DA experiment,
and a simultaneous $PM_{2.5}(PM_{10})$ and lidar AEC DA experiment) are conducted
based on AEC profiles captured by five lidars (located in Beijing,
Shijiazhuang, Taiyuan, Xuzhou, and Wuhu) in the period from 0000 to 1200
UTC, November 13, 2018 as well as MC measurements for $PM_{2.5}$ and $PM_{10}$
taken at over 1,500 ground environmental monitoring stations across China.
DA and forecast results are evaluated based on MC measurements for surface
$PM_{2.5}$. A comparison with the control experiment involving no DA finds that
the 3-DVAR DA system is effective at assimilating lidar AEC data. While
there are only five lidars within the simulation region (approximately 2.33
million $km^2$ in size), assimilating AEC data acquired by these lidar alone can
also effectively improve the accuracy of the initial field and the forecast
performance of the model for $PM_{2.5}$. Moreover, the positive effect resulting
from the optimization of the initial field on forecast performance for $PM_{2.5}$ can
last for more than 24 h. Lidar AEC DA is advantageous because it improves
the accuracy of the vertical $PM_{2.5}MC$ profile. Surface $PM_{2.5}(PM_{10})$ DA is
advantageous because it optimizes the near-surface $PM_{2.5}MC$ distribution.
Furthermore, simultaneous lidar AEC and surface $PM_{2.5}(PM_{10})$ DA can
effectively help integrate their observation information to generate a more
accurate 3D aerosol analysis field.

**Code and data availability:** WRF-Chem model source code can be download
at the WRF model download page (https://www2.mmm.ucar.edu/wrf/users/
download/get_source.html). This 3-DVAR data assimilation system is
developed by ourself. A version of the 3-DVAR code and lidar profile data for



supporting this paper are available at: https://zenodo.org/record/3937564. Full access is available on request from the corresponding author (ywlx_1987@163.com)

**Author contribution:** Yanfei Liang performed numerical experiments, data analysis and wrote the initial manuscript. Yanfei Liang,Zengliang Zang and Wei You developed the 3-DVAR data assimilation system, designed this study and revised the manuscript. Zengliang Zang supervised the project of development. All the authors continuously discussed the 3-DVAR system development and the results of the manuscript.

**Competing interests:** The authors declare that they have no conflict of interest.

**Acknowledgments**

This research was primarily supported by the National Natural Science Foundation of China (Grant No.41775123 and No.41805092), the National Key Research and Development Program of China(Grant No. 2017YFC0209803). We thank the China National Environmental Monitoring Center (CNEMC) for providing $PM_{2.5}$ and $PM_{10}$ data through the website (http://www.cnemc.cn/).

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





Table 1. Numerical experiment schemes

| Experiment | Assimilated data | Assimilation region | Continuous assimilation period | Forecast comparison period |
|---|---|---|---|---|
| Control | N/A | N/A | N/A | 11.13 12:00 –11.14 12:00 |
| DA_PM | $PM_{2.5}+PM_{10}$ | D01/D02 | 11.13 00:00 –11.13 12:00 | 11.13 12:00 –11.14 12:00 |
| DA_Ext | Ext | D01/D02 | 11.13 00:00 –11.13 12:00 | 11.13 12:00 –11.14 12:00 |
| DA_PM_Ext | $PM_{2.5}+PM_{10}+Ext$ | D01/D02 | 11.13 00:00 –11.13 12:00 | 11.13 12:00 –11.14 12:00 |











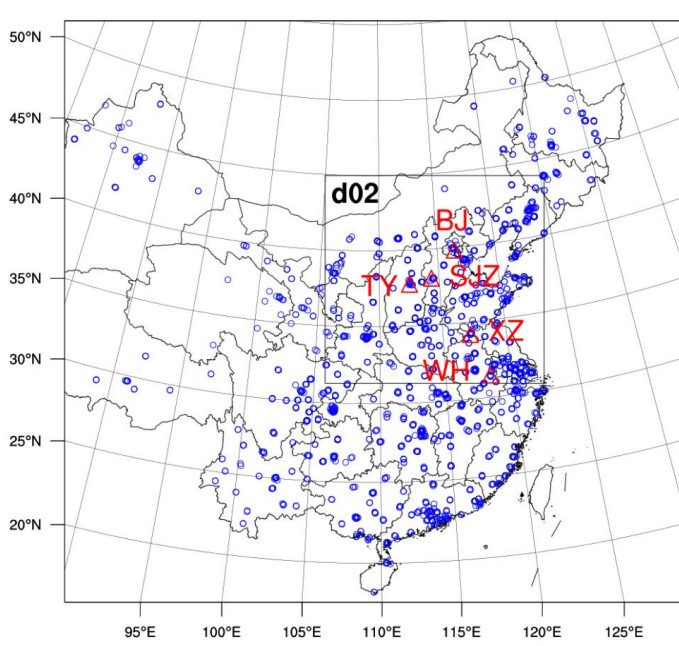


**Figure 1** The double-nested experimental domain. Red triangle and labeling indicate the
locations and names of 5 lidars, and blue circle the locations of 1500 ground environmental
monitoring stations.





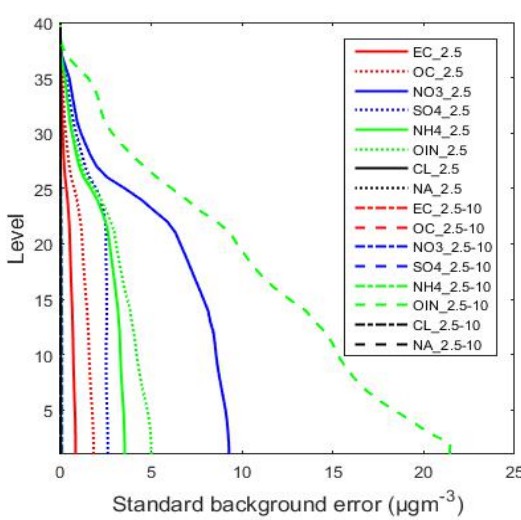


**Figure 2** Vertical BESD profiles of the 16 control variables



















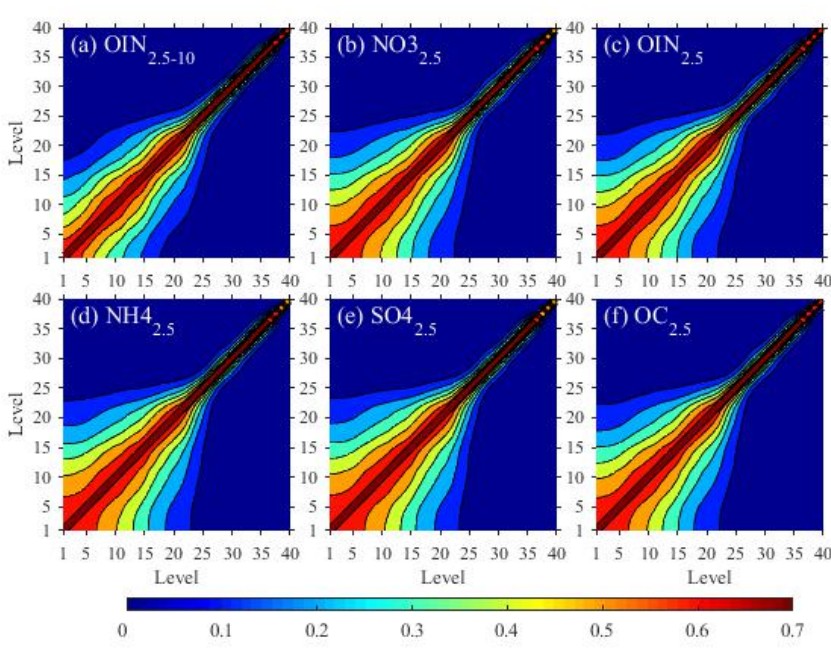


Figure 3 BEVCCs of six control variables

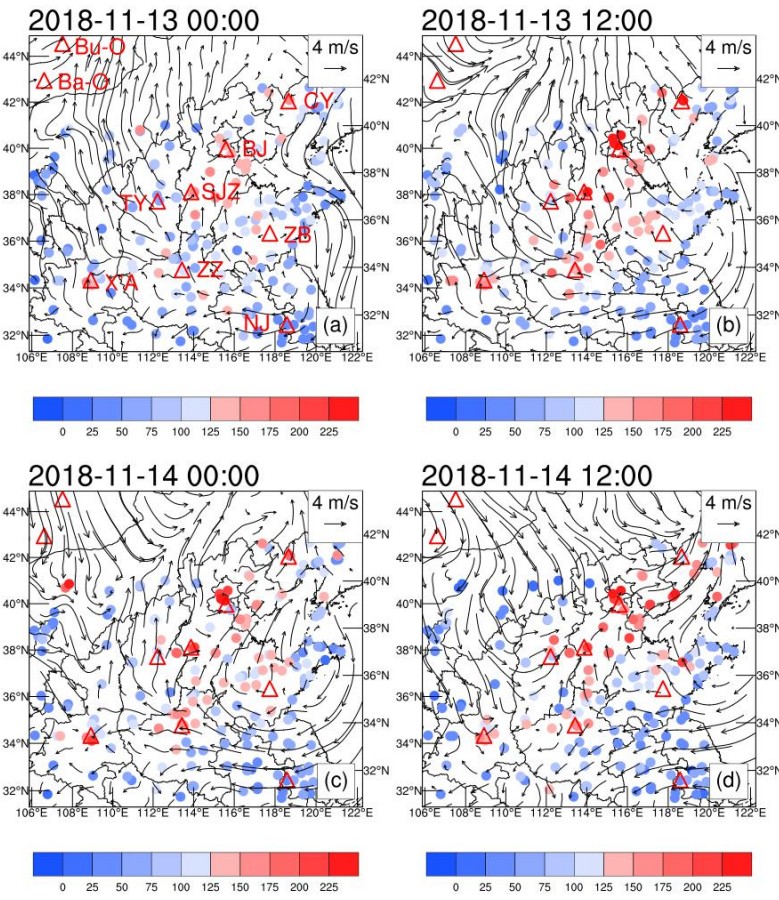


**Figure 4** SurfacePM$_{2.5}$MC measurements in the D02 region and NCEP reanalysis wind field for the period from 0000 UTC, November 13, 2018 to 1200 UTC, November 14, 2018 (CY: Chaoyang; BJ: Beijing; SJZ: Shijiazhuang; TY:Taiyuan; ZB: Zibo; X'A: Xi'an; ZZ: Zhengzhou; NJ: Nanjing; Bu-O: Buyant-Ovoo; Ba-O: Bayan-Ovoo)





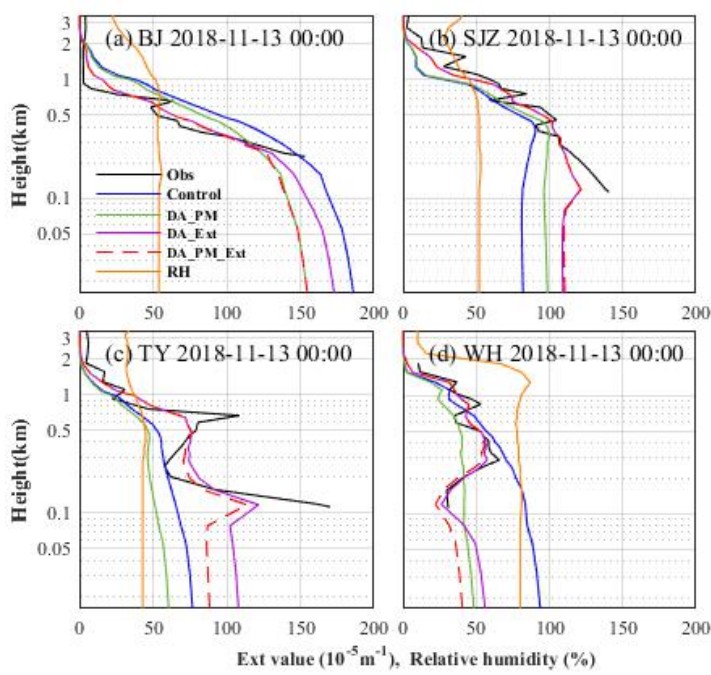

1057

**Figure 5** AEC profiles at 0000 UTC, November 13, 2018 measured at four lidar stations
(black lines) as well as the corresponding AEC profiles obtained from the control (blue
lines) experiment and the DA_PM (green lines), DA_Ext (purple lines) and DA_PM_Ext
(red lines) analysis fields and the simulated RH profiles (orange lines) (BJ: Beijing; SJZ:
Shijiazhuang; TY: Taiyuan; WH: Wuhu)





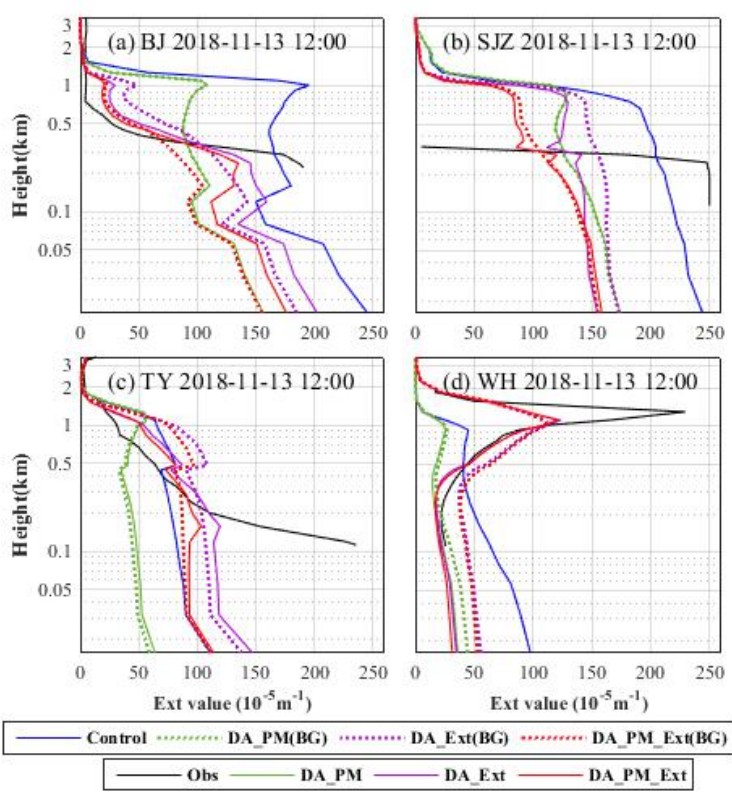

1063

**Figure 6** AEC profiles at 1200 UTC, November 13, 2018 measured at four lidar stations (solid black lines) as well as the corresponding AEC profiles obtained from the control experiment (solid blue lines) and the background (dotted lines) and analysis (solid lines) fields of the DA experiments (BJ: Beijing; SJZ: Shijiazhuang; TY: Taiyuan; WH: Wuhu)




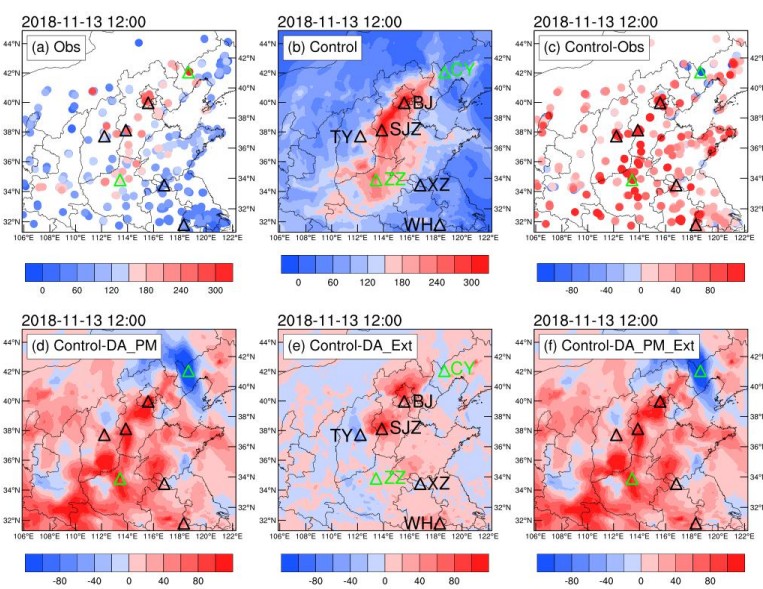

1068

**Figure 7** Surface PM$_{2.5}$MCs measured at 1200 UTC, November 13, 2018 (a), as well as the initial field (b) of the control experiment and its error (c) and the distribution of differences between the initial fields of the control and DA experiments (d, e, and f) (black triangles signify the locations of the lidar stations) (BJ: Beijing; SJZ: Shijiazhuang; TY: Taiyuan; ZZ: Zhengzhou; XZ: Xuzhou; WH: Wuhu)













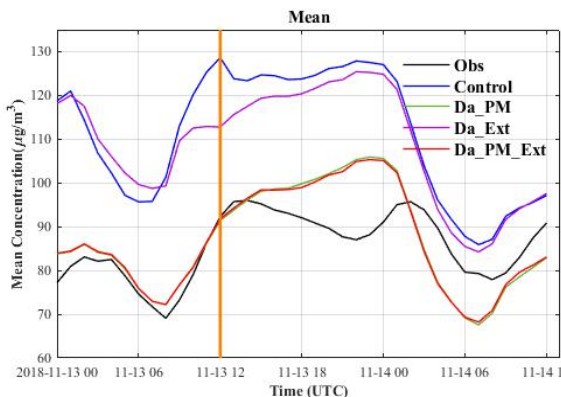


**Figure 8** Variation in the regional mean $PM_{2.5}MC$ with time (the vertical orange line
separates the DA and forecast periods; the black line signifies measurements; the blue line
signifies the $PM_{2.5}MCs$ obtained from the control experiment; the green, purple, and red
lines signify the $PM_{2.5}MCs$ obtained from the DA_PM, DA_Ext, and DA_PM_Ext
experiments, respectively)




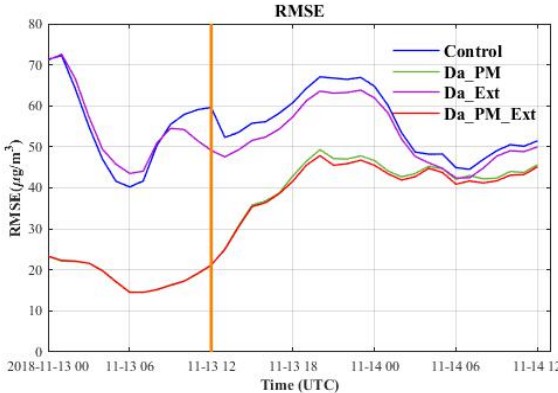


**Figure 9** Variation in the RMSE of surface PM$_{2.5}$MC forecasts with time (the vertical orange line separates the DA and forecast periods; the blue line signifies the PM$_{2.5}$MCs obtained from the control experiment; the green, purple, and red lines signify the PM$_{2.5}$MCs obtained from the DA_PM, DA_Ext, and DA_PM_Ext experiments, respectively)