# Peer review of "Development of a three-dimensional variational assimilation system for lidar profile data based on a size-resolved aerosol model in WRF-Chem model v3.9.1 and its application in PM$_{2.5}$"

_Geoscientific Model Development, 2020_

## Referee Comment (RC1) · Anonymous Referee #1 · 29 Aug 2020

The study of Liang et al. developed a 3-Dvar assimilation system for lidar aerosol extinction coefficient (AEC) data assimilation coupling with WRF-Chem model. To avoid complex Tangent linear /Adjoint development, the IMPROVE algorithm converting PM mass concentration to aerosol extinction coefficient were used. Three assimilation experiments with Lidar AEC, surface PM concentrations, and both Lidar AEC and surface PM were conducted and compared. The analysis and forecast show that PM simulation and forecast skills, especially the vertical profiles were improved in eastern China

with additional AEC assimilation. The publication is well written, clearly structured, and the analyses are comprehensive. I support publication of this manuscript and have only a few small comments and clarification that may require minor revisions.

1. L257-261. It seemed the vertical resolution of Lidar data is much finer than that of the model. Can you add a few words on the uncertainty of the Lidar AEC data? And also clarify how many data were filtered out? Thus the readers may get some more ideas why the complex data preprocess is necessary here.

2. L285-287. It may worth trying to test the different thinning (grid-averaging) approach, from 5x5 to 1x1. As you mentioned that the spatial resolution of the model and the representativeness of Lidar AEC and surface PM data are important, since the inconsistency may cause the adjustments in two directions. It might be interesting to check if no grid-averaging is done before assimilation, but it's only a suggestion for your future study.

3. Section 2.3. It would be nice to add the information of observational errors for AEC and surface PM.

4. L370. Actually the application of IMPROVE algorithm is very important in this study since it simplify the complex adjoint process in the system which is innovative and interesting. However as you discussed, it may bring some uncertainties too (from observed AEC to constrain model species' concentration) since the verification of the IMPROVE parameters hadn't been thoroughly conducted for the locations where Lidar data is provided. Due to different biases between the Mie algorithm in the model and the IMPROVE algorithm in different regions, different assimilation performance may be achieved at different locations. It's suggested to clarify this point more clearly here or in the discussion.

5. L543-546. Does it also indicate different model performances for the vertical profiles at different locations? Or is it related with the different IMPROVE parametrizations for those locations? Some discussion may be nice to help the readers understand more

clearly.

6. L571 Figure 6 -> 7? Please clarify

7. L599. Actually large changes were expected to occur after sunset since PBLH and hence PM concentration change dramatically in a few hours later. For 12UTC (20LST) , it's only 2-3 hours after sunset, thus continuous DA for nocturnal period should be conducted.

---

## Referee Comment (RC2) · Anonymous Referee #2 · 30 Aug 2020

The subject of the article is promising and the results look interesting. Unfortunately the way it is written does not give the article the best output.

The abstract should be rewritten as it is really unclear. 171-175: you should specify it is the EARLINET network. 200-201: you should specify that the aerosol types will be describe later. 392-393: can you write PM10=PM2.5+.... for more clarity. Chapter 3: for each figure you have written "the figure demonstrates", figure can demonstrate

nothing... Also except in the paragraph 3.4, no numbers are given, you just make qualitative comparison. Some more precise results will be welcome. Figure 4: it is not easy to read, maybe you should change the symbol color for the station. Figure 7: what are the green triangles? 691-694: You are doing 2 sentences to repeat the same just with the diurnal specification. You could do it in only one sentences.

The results behind looks interesting but I got a little bit frustrated that you have not been more precised on the results. Can you put some effort on adding some quantitative results (ie. increase by 10%, decrease by 0.2....)

I would like to encourage you to ask an English native to review your article.

---

## Author Comment (AC1) · 5 Oct 2020

Responses to the comments of Reviewer #1: We are truly grateful to yours' positive comments and thoughtful suggestions. Those comments are all valuable and very helpful for revising and improving our paper, as well as the important guiding significance to our researches. Based on these comments and suggestions, we have studied comments carefully and have made correction which we hope meet with approval. All

changes made to the text are marked in red color. Below you will find our point-by-point responses to the reviewers' comments/ questions:

Specific Comments: 1. L257-261. It seemed the vertical resolution of Lidar data is much finer than that of the model. Can you add a few words on the uncertainty of the Lidar AEC data? And also clarify how many data were filtered out? Thus the readers may get some more ideas why the complex data preprocess is necessary here.

Response: We followed the suggestion, and the following information has been added in the revised manuscript (L226-234 and L257-260). The relative standard deviation of the aerosol parameter profiles captured by the lidar over Beijing was 20.4% in the height range of 1-2 km. This lidar was calibrated via comparative observation of several lidars (Chen et al., 2019). The precision of the AEC profiles released by the other four lidars was below the quality margins (25% of the typical AEC observed in the planetary boundary layer or $\pm0.01km-1$), as defined by Matthias et al. (2004). However, the relative standard deviation of the aerosol parameter profiles in the height range of 2-5 km released by lidar over Beijing was 35.9%. After the quality control process, 84.32% of the original AEC data from the lidar over Beijing were accepted as valid data, and 88.75%, 54.10%, 26.74%, and 10.95% of the data from the Taiyuan, Wuhu, Shijiazhuang, and Xuzhou lidars, respectively, were valid.

2. L285-287. It may worth trying to test the different thinning (grid-averaging) approach, from 5×5 to 1×1. As you mentioned that the spatial resolution of the model and the representativeness of Lidar AEC and surface PM data are important, since the inconsistency may cause the adjustments in two directions. It might be interesting to check if no grid-averaging is done before assimilation, but it's only a suggestion for your future study.

Response: We really appreciate your valuable suggestion. Actually, the scale of averaging observation data is one of the important parameters that we need to determine. More detail please see the supplemental file.

3. Section 2.3. It would be nice to add the information of observational errors for AEC and surface PM.

Response: Thank you for your suggestion. Please see the supplemental file.

4. L370. Actually the application of IMPROVE algorithm is very important in this study since it simplify the complex adjoint process in the system which is innovative and interesting. However as you discussed, it may bring some uncertainties too (from observed AEC to constrain model species' concentration) since the verification of the IMPROVE parameters hadn't been thoroughly conducted for the locations where Lidar data is provided. Due to different biases between the Mie algorithm in the model and the IMPROVE algorithm in different regions, different assimilation performance may be achieved at different locations. It's suggested to clarify this point more clearly here or in the discussion.

Response: We really appreciated and followed the suggestion, and have added the following words in the revised manuscript (L763-769). On the one hand, datasets from which the IMPROVE parameters were determined in previous studies were measured in specific regions and near the ground. The verification of the IMPROVE parameters had not been thoroughly conducted for the locations where lidar data were provided. Therefore, there may have been different biases between the Mie algorithm and the IM-PROVE algorithm in different regions, inducing inconsistent assimilation performance.

5. L543-546. Does it also indicate different model performances for the vertical profiles at different locations? Or is it related with the different IMPROVE parametrizations for those locations? Some discussion may be nice to help the readers understand more clearly.

Response: We are so sorry for that the description in L543-546 is not clear enough, which increases reading difficulties for readers. What we are concerned about here is that while the lidar data are not available at surface, the DA_Ext could adjust the surface PM MCs significantly, but the adjustments could not always have positive effect. The

effects of the different model performances and the different IMPROVE parametrizations at different locations are also discussed in chapter 4. The following words have been added in the revised manuscript. Please see the supplemental file.

6. L571 Figure 6 -> 7? Please clarify.

Response: We have revised the legend, notes, and clarified the description of the content, hoping that it will make the article clearer for readers to read.

7. L599. Actually large changes were expected to occur after sunset since PBLH and hence PM concentration change dramatically in a few hours later. For 12UTC (20LST), it's only 2-3 hours after sunset, thus continuous DA for nocturnal period should be conducted.

Response: The characteristics of PBLH and hence PM concentration changes provide us with an important reference for design the applied assimilation scheme. The following words have been added in the revised manuscript. In addition, because the 1200UTC (2000LST) was only 2-3 h after sunset, so large changes of PM concentration profile may occur due to large changes in the PBLH after sunset.

We would like to express our great appreciation to you for the valuable and pertinent comment on our manuscript, which is crucial to improve the quality of our work. We hope that these revisions are satisfactory and that the revised version will be acceptable for publication in Geoscientific Model Development. Thank you very much for your work concerning my paper.

Please also note the supplement to this comment:
https://gmd.copernicus.org/preprints/gmd-2020-223/gmd-2020-223-AC1-supplement.pdf

**Supplement:**

**Responses to the comments of Reviewer #1:**

We are truly grateful to yours' positive comments and thoughtful suggestions. Those comments are all valuable and very helpful for revising and improving our paper, as well as the important guiding significance to our researches. Based on these comments and suggestions, we have studied comments carefully and have made correction which we hope meet with approval. All changes made to the text are marked in red color. Below you will find our point-by-point responses to the reviewers' comments/ questions:

**Specific Comments:**

*1.L257-261. It seemed the vertical resolution of Lidar data is much finer than that of the model. Can you add a few words on the uncertainty of the Lidar AEC data? And also clarify how many data were filtered out? Thus the readers may get some more ideas why the complex data preprocess is necessary here.*

**Response:**

We followed the suggestion, and the following information has been added in the revised manuscript (L226-234 and L257-260). The relative standard deviation of the aerosol parameter profiles captured by the lidar over Beijing was 20.4% in the height range of 1-2 km. This lidar was calibrated via comparative observation of several lidars (Chen et al., 2019). The precision of the AEC profiles released by the other four lidars was below the quality margins (25% of the typical AEC observed in the planetary boundary layer or $\pm 0.01 km-1$), as defined by Matthias et al. (2004). However, the relative standard deviation of the aerosol parameter profiles in the height range of 2-5 km released by lidar over Beijing was 35.9%.

After the quality control process, 84.32% of the original AEC data from the lidar over Beijing were accepted as valid data, and 88.75%, 54.10%, 26.74%, and 10.95%

of the data from the Taiyuan, Wuhu, Shijiazhuang, and Xuzhou lidars, respectively, were valid.

*2.L285-287. It may worth trying to test the different thinning (grid-averaging)*

*approach, from 5×5 to 1×1. As you mentioned that the spatial resolution of the*

*model and the representativeness of Lidar AEC and surface PM data are*

*important, since the inconsistency may cause the adjustments in two directions. It*

*might be interesting to check if no grid-averaging is done before assimilation, but*

*it's only a suggestion for your future study.*

**Response:**

We really appreciate your valuable suggestion. Actually, the scale of averaging observation data is one of the important parameters that we need to determine.

However, no relevant theoretical basis has been found so far. It can only be determined roughly based on experience and a few ideal experiments. In an ideal experiment we designed, the background field is set to 0, the observation error is set to 4.6, and the two observations whose absolute value is slightly larger than the observation error a=-5.0 and b=5.0 are separated by 0.97 grid distances and are within the same grid cell. We believe that the model can only effectively simulate fluctuations with wavelengths greater than twice the grid distance. Therefore, the difference between observation a and observation b within the same grid cell represents random error, and the true value near the grid cell where the two observation points are located should be around 0. After assimilating these two observations, as showed in the following picture, the increments near observation points a and b are close to 0, which is reasonable. However, there is a negative increment center appearing at A at the 7 grid distances to the left of observation point a, and a positive increment center appearing at B at 7 grid distances to the right of observation point b, with the distance of AB reaches 14 grids distance, which is unreasonable. To avoid this unreasonable result, the simple way is averaging the two observations as one before assimilation. From the ideal experiment, we believe that the grid-averaging for observations are necessary before assimilation. As for how to choose the optimal average scale, more researches are needed in the future.

[Figure]

*3.Section 2.3. It would be nice to add the information of observational errors for*

*AEC and surface PM.*

**Response:**

Thank you for your suggestion. First of all, please allow us to introduce the way of calculating observation error covariance matrix appeared in articles we have read.

Following Elbern et al. [2007], Schwartz et al. [2012] and Jiang et al. [2013], the observation error covariance matrix is assumed to be diagonal, that is, the observation errors are not correlated, and the diagonal elements of R ($\varepsilon_{obs}$) are included contributions from measurement errors $\varepsilon_m$ and representation errors $\varepsilon_r$. Elbern et al.

[2007] calculated the $\varepsilon_{obs}= \varepsilon_m+ \varepsilon_r$, whereas Schwartz et al. [2012] and Jiang et al.

[2013] defined the $\varepsilon_{\text{obs}} = \sqrt{\varepsilon_{\text{m}}^2 + \varepsilon_{\text{r}}^2}$ . All the three articles calculated representation errors

$\varepsilon_r$ as $\varepsilon_{\text{r}} = \gamma \varepsilon_m \sqrt{\dfrac{\Delta x}{L}}$ where $\gamma$ is an adjustable parameter scaling $\varepsilon_{\text{m}}$, $\Delta x$ is the grid spacing and L is the radius of influence of an observation. For the $\varepsilon_{\text{m}}$ of PM$_{2.5}$ or PM$_{10}$,

Pagowski et al. [2010] used a PM2.5 measurement error of 2 μg/m3, whereas

Schwartz et al. [2012] and Jiang et al. [2013] used a measurement error defined as

$\varepsilon_{\text{m}} = 1.5 + 0.0075 \times \Pi o$ where $\Pi o$ denotes PM observational values (units: μg/m3). For the $\varepsilon_{\text{m}}$ of AEC, Yumimoto et al. [2008] introduced a minimal absolute error and defined the observation errors $\varepsilon_{\text{m}}$ as $\varepsilon_{\text{m}} = \max(\varepsilon_{\text{abs}}, \Pi o \times \varepsilon_{\text{rel}})$, where $\varepsilon_{\text{abs}}$ represents a minimal absolute error set as 0.05 km$^{-1}$ , $\Pi o$ denotes AEC observational values (units:

km$^{-1}$) and $\varepsilon_{\text{rel}}$ represents the relative error rate, which was assigned as 10%.

Second, please allow us to explain why the information of observational errors is not introduced in the article. The focus of this article is to accomplish the assimilation of AEC by establishing the AEC observation operator, verify the feasibility of the assimilation scheme and find some factors that may affect the assimilation effect.

Because the influence of observation error on the assimilation effect is theoretically predictable, that is, the smaller the observation error, the greater the absolute value of the assimilation incremental field are, and the closer the assimilation analysis field are to the observation field deviating from the background field. In other words, no matter how large the observation error is, as long as the observation operator is correct, the assimilation analysis field will always fall between the background field and the observation field and has a positive assimilation effect, even though not the best.

Because reaching the best assimilation effect through the adjustments of observation error is not the focus of this article, so in order to find factors that may affect the assimilation effect other than observation error, we set the observation error as a constant in the experiment, which is about 50% of the standard deviation of the background error of $PM_{2.5}$ (or $PM_{10}$, AEC). As showed in Section 2.4, the background error standard deviations of the 16 control variables have been calculated by the NMC

method, and the observation operator in Section 2.5 defined the formula between the control variables and $PM_{2.5}$ (or $PM_{10}$, AEC), then by assuming that the background error of the control variables are uncorrelated, the background error standard deviation of $PM_{2.5}$, $PM_{10}$ and AEC can be obtained. The observational errors of $PM_{2.5}$,

$PM_{10}$ and AEC used in this article are $5.80 \mu g/m^3$, $12.18 \mu g/m^3$ and $0.01 km^{-1}$, respectively.

4.*L370. Actually the application of IMPROVE algorithm is very important in this*

*study since it simplify the complex adjoint process in the system which is*

*innovative and interesting. However as you discussed, it may bring some*

*uncertainties too (from observed AEC to constrain model species' concentration)*

*since the verification of the IMPROVE parameters hadn't been thoroughly*

*conducted for the locations where Lidar data is provided. Due to different biases*

*between the Mie algorithm in the model and the IMPROVE algorithm in different*

*regions, different assimilation performance may be achieved at different locations.*

*It's suggested to clarify this point more clearly here or in the discussion.*

**Response:**

We really appreciated and followed the suggestion, and have added the following words in the revised manuscript (L763-769).

On the one hand, datasets from which the IMPROVE parameters were determined in previous studies were measured in specific regions and near the ground.

The verification of the IMPROVE parameters had not been thoroughly conducted for the locations where lidar data were provided. Therefore, there may have been different biases between the Mie algorithm and the IMPROVE algorithm in different regions, inducing inconsistent assimilation performance.

5.*L543-546. Does it also indicate different model performances for the vertical*

*profiles at different locations? Or is it related with the different IMPROVE*

*parametrizations for those locations? Some discussion may be nice to help the*

*readers understand more clearly.*

**Response:**

Thank you very much for your suggestion. We are so sorry for that the description in L543-546 is not clear enough, which increases reading difficulties for readers. What we are concerned about here is that while the lidar data are not available at surface, the DA_Ext could adjust the surface PM MCs significantly, but the adjustments could not always have positive effect. The effects of the different model performances and the different IMPROVE parametrizations at different locations are also discussed in chapter 4.

The following words have been added in the revised manuscript.

L525-536: The DA increments of AEC values from the DA_PM, that is, the AEC

values obtained from the DA_PM experiment (green lines) minus those from the control experiment (blue lines), were negative for Beijing (Figure 5a), Taiyuan (Figure 5c), and Wuhu (Figure 5d) at the surface. They were also negative from the near-surface to a height of about 1000 m, although their absolute values were smaller than those at the surface. This is because the BEVCCs between each in-air layer and the surface layer were positive and decreased with height (Figure 3), so that the information contained in the surface PM MC measurements was spread to the air.

However, the results of the adjustment of the AEC profiles were not always positive, because the aerosol bias of the control experiment at the surface was not always the same as it was in the atmosphere.

L546-552: In addition, although lidar data were not available at the surface, the

DA_Ext adjusted of the surface PM MCs, corrected the overestimation of surface

$PM_{2.5}$MCs in Beijing and Wuhu, but increased the overestimation of surface

$PM_{2.5}$MCs in Taiyuan. This is because the information contained in the in-air AEC

was spread to the surface, while the aerosol bias of the control experiment in the air did not always match that at the surface.

6.L571 *Figure 6 -> 7? Please clarify.*

**Response:**

We are so sorry for that the description legend, notes, and the description of the content shown in the figure 6 and figure 7 are not clear enough. We have revised the legend, notes, and clarified the description of the content, hoping that it will make the article clearer for readers to read.

7.L599. *Actually large changes were expected to occur after sunset since PBLH and hence PM concentration change dramatically in a few hours later. For 12UTC (20LST), it's only 2-3 hours after sunset, thus continuous DA for nocturnal period should be conducted.*

**Response:**

Thank you very much for your opinion. The  characteristics of PBLH and hence PM concentration changes provide us with an important reference for design the applied assimilation scheme. The following words have been added in the revised manuscript (L603-606).

In addition, because the 1200UTC (2000LST) was only 2-3 h after sunset, so large changes of PM concentration profile may occur due to large changes in the PBLH after sunset.

We would like to express our great appreciation to you for the valuable and pertinent comment on our manuscript, which is crucial to improve the quality of our work. We hope that these revisions are satisfactory and that the revised version will be acceptable for publication in Geoscientific Model Development. Thank you very much for your work concerning my paper.

Wish you all the best!

Yours sincerely,

Yanfei Liang, Wei You and Zengliang Zang

05/10/2020

---

## Author Comment (AC2) · 5 Oct 2020

Responses to the comments of Reviewer #2: We are truly grateful to yours' positive comments and thoughtful suggestions. Those comments are all valuable and very helpful for revising and improving our paper, as well as the important guiding significance to our researches. Based on these comments and suggestions, we have studied comments carefully and have made correction which we hope meet with approval. All

changes made to the text are marked in blue color. Below you will find our point-by-point responses to the reviewers' comments/ questions:

Specific Comments:

1. The abstract should be rewritten as it is really unclear.

Response: We followed the suggestion, and the abstract has been rewritten as following in the revised manuscript.

2. L171-175: you should specify it is the EARLINET network. L200-201: you should specify that the aerosol types will be described later. L392-393: Can you write PM10=PM2.5+... for more clarity. Chapter 3: for each figure you have written "the figure demonstrates", figure can demonstrate nothing...

Response: We are truly grateful to your thoughtful suggestions and changes in the revised manuscript are as following: In L169-175, we have specified that the data are "captured by 12 lidars positioned in the Mediterranean Basin from the ACTRIS (Aerosols, Clouds, and Trace Gases Research InfraStructure)/EARLINET (European Aerosol Research Lidar Network) and one lidar positioned on the French island of Corsicain from the framework of the pre-ChArMEx (Chemistry-Aerosol Mediterranean Experiment)/TRAQA (TRansport àlongue distance et Qualité de l'Air)." In L201-203, we have specified that "This scheme, which will be described in Section 2.4, can be used to predict the profiles of eight aerosol types." In L403-404, we have write that PM10=PM2.5+SO42.5-10+NO32.5-10+NH42.5-10+OC2.5-10+EC2.5-10+CL2.5-10 +NA2.5-10+OIN2.5-10 The expression "the figure demonstrates" have been removed or replaced by "as showed in figure".

3. Except in the paragraph 3.4, no numbers are given, you just make qualitative comparison. Some more precise results will be welcome.

Response: We really appreciate this suggestion and follow the suggestion. We have added more quantitative results in the Abstract section (L45-53) and Conclusion sec-
tion (L795-801)

4. Figure 4: It is not easy to read, may be you should change the symbol color for the station.

Response: The symbol color for the station has been changed to black and the line of wind vector and the map province boundary has been set thinner in the revised manuscript.

5. Figure 7: What are the green triangles?

Response: We are so sorry for that our lack of clear description of the mark in figure 7 has troubled readers. These two green triangles mark the locations of the two cities mentioned in the description for figure 7 but without lidar. We have added "green triangles mark the locations of the two cities without lidar " in the revised manuscript.

6. L691-694: You are doing 2 sentences to repeat the same just with the diurnal specification. You could do it in only one sentence.

Response: We followed the suggestion. The original expression has been changed in L646-648 as " Figure 8 shows the variation of the regional mean of the PM2.5MC over time from the four experiments. The regional mean of the PM2.5MC (black line) exhibited a notable diurnal pattern." Redundant expressions similar to this have also been changed in the revised manuscript.

7. The results behind looks interesting but I got a little bit frustrated that you have not been more precise on the results. Can you put some effort on adding some quantitative results (ie. increase by 10%, decrease by 0.2. . .. ).

Response: We have added more quantitative results in the Abstract and Conclusion section(L45-53 and L795-801). Also, please allow us to explain why few quantitative results are introduced in the article except in the paragraph 3.4. Firstly, the quantitative analysis of direct effects of DA in the paragraph 3.3 have been given in paragraph 3.4, as the end of DA period is the initial time of forecast period. In addition, the focus of this

article is to accomplish the assimilation of AEC by establishing the AEC observation operator, verify the feasibility of the assimilation scheme and find some factors that may affect the assimilation effect. And to what extent the assimilation improves the forecasting effect are not what we trying to emphasize.

8. I would like to encourage you to ask an English native to review your article.

Response: We followed the suggestion. We have carefully revised the manuscript. In addition, we have asked a freelance English editor to improve the presentation.

We would like to express our great appreciation to you for the valuable and pertinent comment on our manuscript, which is crucial to improve the quality of our work. We hope that these revisions are satisfactory and that the revised version will be acceptable for publication in Geoscientific Model Development. Thank you very much for your work concerning our paper.

Please also note the supplement to this comment:
https://gmd.copernicus.org/preprints/gmd-2020-223/gmd-2020-223-AC2-supplement.pdf